# Exploring Image Augmentations for Siamese Representation Learning with Chest X-Rays

**Rogier van der Sluijs**[*1]               SLUIJS@STANFORD.EDU
**Nandita Bhaskhar**[*2]               NANBHAS@STANFORD.EDU
**Daniel L Rubin**[1,3,4]               DLRUBIN@STANFORD.EDU
**Curtis P Langlotz**[1,3,5]             LANGLOTZ@STANFORD.EDU
**Akshay S Chaudhari**[1,3]             AKSHAYSC@STANFORD.EDU

[1] *Department of Radiology, Stanford University*

[2] *Department of Electrical Engineering, Stanford University*

[3] *Department of Biomedical Data Science, Stanford University*

[4] *School of Medicine, Stanford University*

[5] *Department of Biomedical Informatics Research, Stanford University*

**Editors:** Accepted for publication at MIDL 2023

## Abstract

Image augmentations are quintessential for effective visual representation learning across self-supervised learning techniques. While augmentation strategies for natural imaging have been studied extensively, medical images are vastly different from their natural counterparts. Thus, it is unknown whether common augmentation strategies employed in Siamese representation learning generalize to medical images and to what extent. To address this challenge, in this study, we systematically assess the effect of various augmentations on the quality and robustness of the learned representations. We train and evaluate Siamese Networks for abnormality detection on chest X-Rays across three large datasets (MIMIC-CXR, CheXpert and VinDr-CXR). We investigate the efficacy of the learned representations through experiments involving linear probing, fine-tuning, zero-shot transfer, and data efficiency. Finally, we identify a set of augmentations that yield robust representations that generalize well to both out-of-distribution data and diseases, while outperforming supervised baselines using just zero-shot transfer and linear probes by up to 20%. Our code is available at `https://github.com/StanfordMIMI/siaug`.

**Keywords:** Data Augmentations, Self-Supervised Learning, Medical Imaging.

## 1. Introduction

Deep learning algorithms enable high-accuracy medical image analysis, yet are constrained by limitations of labelled data. Determining ground-truth image labels for diagnostic and prognostic tasks typically involves multiple annotators with clinical expertise and is often costly, time-consuming, and subject to inter-reader variability (Kim et al., 2022). Such a scarcity of annotated datasets has spurred research in data-efficient deep learning techniques, such as transfer learning and self-supervision (Krishnan et al., 2022). ImageNet pretraining is common, yet transferring representations from natural images is not always

---

[*] Contributed equally

successful, possibly due to the shifted distribution and visual features of medical images (Raghu et al., 2019). Self-supervision, on the other hand, exploits the intrinsic structure of unlabelled data to learn effective representations, which can then be used for fine-tuning or zero-shot transfer on downstream tasks. Self-supervision proves to be particularly useful in medicine, given the abundance of unlabelled imaging data. It also provides robustness to out-of-distribution data (Hendrycks et al., 2019) and concept drifts. Learning visual features without a strong supervisory signal, however, is challenging.

One particularly powerful technique used in self-supervision is to compare two or more augmented views of the same image using a Siamese network architecture (Bromley et al., 1993). A common denominator among variants of this technique, such as contrastive learning (Chen et al., 2020a; He et al., 2020) and feature prediction (Grill et al., 2020; Caron et al., 2021; Chen and He, 2021), is their reliance on an augmentation strategy to generate different views of the input data. The question "what makes good views" has been explored in-depth for natural images in the context of contrastive learning (Tian et al., 2020; Chen et al., 2020a), but has not been answered for medical tasks. Efforts to transfer common augmentation strategies to pretrain representations on medical data have thus far had limited success compared with hand-crafted strategies (Azizi et al., 2021; Sowrirajan et al., 2021).

To address these limitations, we systematically evaluate the effectiveness, robustness, and generalizability of image augmentation strategies for representation learning on three large datasets of chest x-rays (Irvin et al., 2019; Johnson et al., 2019; Nguyen et al., 2022). In this study, we assess an extensive range of augmentations through linear probing, zero-shot transfer, fine-tuning, and data efficiency experiments and show that:

- Visual representations extracted with different augmentations results in substantial variations on downstream classification tasks (up to 18% difference). Random resized cropping largely defines optimal performance of the learned representations on downstream tasks.

- Representations learned with the optimal set of augmentations outperform supervised baselines on several occasions on both internal (by 13.6-20.0%) and external evaluation (up to 27.0%) sets.

- Zero-shot transfer, linear probing, and fine-tuning with limited data using pretrained representations surpass classification accuracy of their supervised counterparts on several occasions.

- The learned features are robust to forms of label drift and catastrophic forgetting, and show success in classification of diseases that are rare (e.g. Rib Fractures [RF]) and unseen across datasets (e.g. Tuberculosis [TB]).

## 2. Related Work

**Self-supervised learning**. Self-supervision typically involves formulating a pretext task solely to learn a good representation of the data. This representation can subsequently be fine-tuned on a downstream task in a data-efficient manner. A broad range of such pretext

tasks exist, such as solving jigsaw puzzles (Noroozi and Favaro, 2016; Taleb et al., 2021), image rotation prediction (Gidaris et al., 2018), and context restoration (Pathak et al., 2016).

**Contrastive learning**. Contrastive visual representation learning seeks to contrast positive pairs of image views with negative pairs (Hadsell et al., 2006). Positive pairs are created from the input data, whereas negative pairs are sampled from a mini-batch (Chen et al., 2020a) or queue (Chen and He, 2021). Traditional contrastive learning requires positive pairs and a large sample of negative pairs for effective training. Variations of contrastive methods use approaches that do not rely on negative pairs. BYOL (Grill et al., 2020) introduced a Siamese network trained to predict views of opposing branches. Extensions of this framework explore different architectural components, such as the loss function, projection heads, and the teacher-student architecture (Caron et al., 2021; Chen and He, 2021).

**Image augmentations for self-supervision**. Data augmentations are widely used in supervised learning to increase the diversity of the training data and to improve generalizability (Krizhevsky et al., 2017; Cubuk et al., 2018). RandAugment (Cubuk et al., 2020) is a powerful method that applies a randomly selected subset of predefined augmentations to the input data. In contrast, in self-supervised learning, augmentations are often applied to construct a pretext task (Tian et al., 2020). Common augmentations for contrastive learning were explored in SimCLR (Chen et al., 2020a). In the medical domain, amongst others, affine transformations, elastic deformations (Chaitanya et al., 2020), and physics-driven augmentations (Desai et al., 2022) have been considered for self-supervised learning.

**Self-supervised learning for Chest X-Rays**. Chest X-Ray classification is a well-studied subject, and its recent role has been amplified in light of the COVID-19 pandemic (Wynants et al., 2020). Self-supervision has emerged as viable strategy to aid the detection of pathologies on chest x-rays (Gazda et al., 2021; Azizi et al., 2022). Multi-modal vision-language learning has shown to be effective (Zhang et al., 2020; Huang et al., 2021; Tiu et al., 2022; Delbrouck et al., 2022), but necessitates the availability of radiology reports. The current study is most closely aligned with the image-only augmentation strategies examined in MoCo-CXR (Sowrirajan et al., 2021), MICLe (Azizi et al., 2021), and REMEDIS (Azizi et al., 2022). These studies, however, use contrastive methods that rely on negative sampling and were not designed to systematically explore augmentation strategies.

## 3. Methods

To evaluate the impact of data augmentations on the quality of the learned representations, we used SimSiam (Chen and He, 2021) - a minimal Siamese network architecture. SimSiam does not rely on negative sampling, knowledge distillation, or prototype clustering, which allows us to most directly study the role of augmentations in Siamese representation learning.

### 3.1. Architecture and Pretraining Objective

The architecture of SimSiam consists of two identical and weight-sharing branches that each take an augmented view (i.e. $x_1$ and $x_2$) of the same image $x$ as an input (Figure 1). Both views ($x_1$ and $x_2$) are processed by an identical encoder network, $f(\cdot)$, that outputs

feature vectors $f(x_i)$. These feature vectors are passed on to a two-layered MLP projector network $g(\cdot)$ that produces a low-dimensional latent representation $z_i = g(f(x_i))$ of the data.

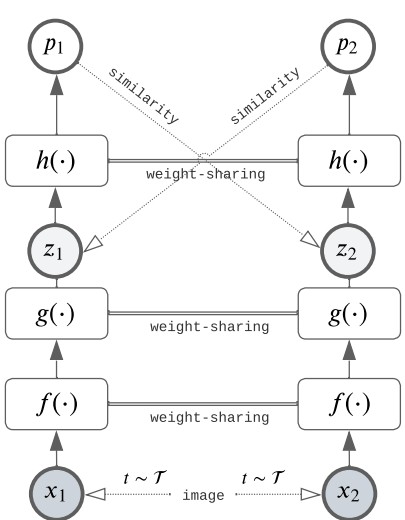

Figure 1: SimSiam architecture.

As a final step, the latent representations produced by each branch $(z_i)$ are input to a predictor network $h(\cdot)$. The predictor network is an MLP that aims to predict the projection $z$ of the opposing branch (i.e. $h_1(z_1) = p_1$ tries to predict $z_2$, while $h_2(z_2) = p_2$ tries to predict $z_1$). The loss function $\mathcal{L}$ is defined as the negative cosine similarity between the predictions of the predictor networks $p_1$ and $p_2$ and the actual projected feature vectors $z_1$ and $z_2$:

$$\mathcal{L} = -\frac{1}{2}\left(\frac{p_1}{\|p_1\|_2} \cdot \frac{z_2}{\|z_2\|_2}\right) - \frac{1}{2}\left(\frac{p_2}{\|p_2\|_2} \cdot \frac{z_1}{\|z_1\|_2}\right) \quad (1)$$

where $\|\cdot\|_2$ is the $l_2$ norm. Note that, unlike typical contrastive self-supervised learning, calculation of the loss does not involve negative samples.

## 3.2. Data Collection

Frontal chest x-rays from three publicly available datasets were used to train and evaluate our models. First, the MIMIC-CXR (Johnson et al., 2019) dataset (from Boston, USA) includes images acquired from 277,835 imaging studies of patients, of which $n = 200,000$ images were used for training and validation, and $n = 37,962$ were used for evaluation. Second, the CheXpert (Irvin et al., 2019) dataset (from Stanford, USA) contains chest x-rays from 65,240 patients, of which $n = 168,660$ images were used for training and validation, and $n = 22,367$ were used for evaluation. In both CheXpert and MIMIC-CXR, an automatic radiology report labeller (Irvin et al., 2019) was used to annotate each report/image pair for the presence of 14 different conditions of which a diverse subset was included (Appendix A, Table 5). Third, the VinDr-CXR (Nguyen et al., 2022) dataset (from Vietnam) contains 18,000 chest x-rays of which 15,000 were each manually labelled by three radiologists for 22 critical findings and 6 diagnoses in the training set. Every evaluation set image ($n = 3,000$) was annotated by five radiologists. The sophisticated labelling makes VinDr-CXR an optimal dataset for evaluation purposes (Appendix A, Table 6).

## 3.3. Experimental Setup and Study Design

**Training Pipeline**. Our training pipeline consists of (i) self-supervised pretraining of an encoder, $ResNet(\cdot)$, using unlabelled images via SimSiam (Section 3.1), (ii) supervised linear probing (i.e. training a single-layer classifier on top of a frozen encoder), and (iii) supervised fine-tuning of the entire encoder initialized with the weights of a pretrained encoder and a pretrained classification head.

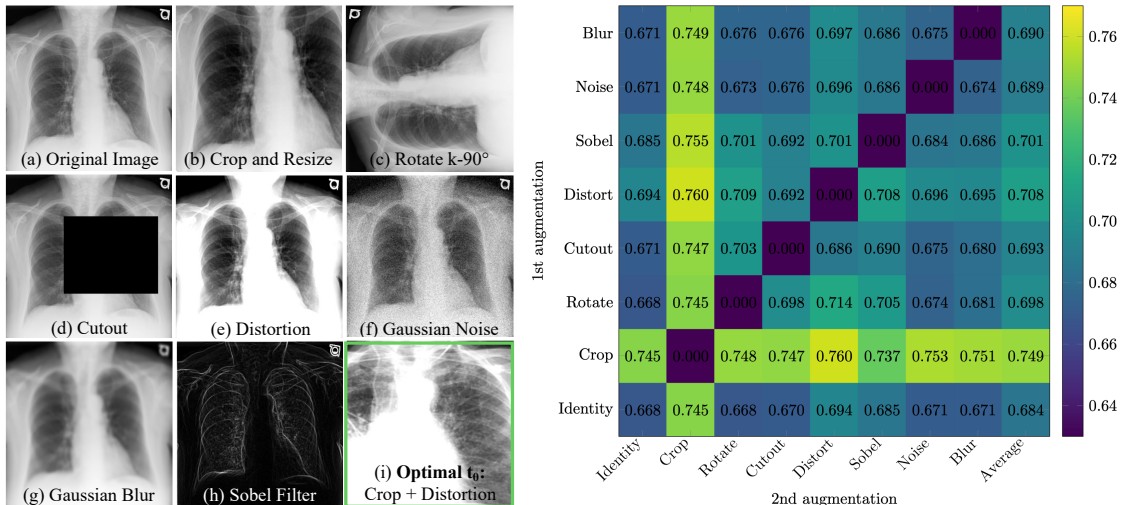

Figure 2: A selection of image augmentations (left: a-h) and their performance on MIMIC-CXR evaluation data (right), following pretraining and linear probing on MIMIC-CXR training data. Combining random resized cropping with distortion resulted in the best augmentation pair, $t_\theta$ .

**Datasets**. We use the unlabelled MIMIC-CXR training split for all self-supervised pretraining experiments. We provide pretraining results with CheXpert in Appendix F.1 (Tables 10 and 11). We perform supervised linear probing and fine-tuning on labelled train splits of MIMIC-CXR, CheXpert, and VinDr-CXR. For evaluation, we use held out data from the internal evaluation split (MIMIC-CXR) as well as external evaluation splits (CheXpert and VinDr-CXR). All dataset splits and the included labels are detailed in Appendix A (Tables 5 and 6). The three datasets encompass multi-site data and include different diseases and occurrence rates. Some labels are overlapping, while others are unseen in the pretraining data (e.g. tuberculosis [TB] in VinDr-CXR), forming a test-bed for comprehensive evaluations.

**Training Details**. We use the SimSiam architecture (Chen and He, 2021) with a ResNet-50 encoder (He et al., 2016) for all experiments involving representation learning. The SGD optimizer (with $weightdecay = 0.0001$ and $momentum = 0.9$) is used for pretraining ($lr = 0.05$), linear probing ($lr = 30$, $weight\ decay = 0$), and fine-tuning ($lr = 0.00001$); using a cosine decay learning rate scheduler (Chen and He, 2021). Batch sizes were fixed to 256. Experiments were trained with PyTorch on 8 NVIDIA A100 GPUs on a single node with 32-bit floating point precision.

**Evaluation Metrics**. We evaluate the quality of our pretrained representations by measuring their downstream discriminative performance (averaged and per label), generalization capability, and data efficiency using the following multi-label metrics: (i) Macro AUROC (area under receiver operating curve), (ii) label-wise AUROC, (iii) Hamming Loss, and (iv) Ranking Error (Tsoumakas et al., 2010) for holistic evaluation. We report Macro AUROC in the main manuscript and the rest, along with their descriptions and motivation in Appendix E, F.

## 4. Experiments and Findings

### 4.1. Optimal Augmentation Strategy

We seek to learn invariant features from augmented image views during pretraining. Inspired by the systematic study of augmentations for SimCLR by Chen et al. (2020a), we first explore the efficacy of augmentations in isolation. We evaluate three common geometric/spatial transformations, namely resized cropping, rotation (Gidaris et al., 2018), and cutout (DeVries and Taylor, 2017), along with pixel-wise transformations of distortion (i.e. brightness/contrast), Gaussian noise, Gaussian blur, and Sobel filtering (Figure 2).

First, we apply the identity transformation to one branch of the Siamese network, and apply a single augmentation $t_i \in \mathcal{T}$ to the other branch (i.e. $t_1(x_i)$). We repeat this procedure with pairs of augmentations (i.e. $t_2(t_1(x_i))$) as shown in Figure 2. We pretrain our models on MIMIC-CXR and evaluate their performance based on supervised linear probing (Zhang et al., 2016) on the MIMIC-CXR validation set. We refer to the pair of augmentations with the highest Macro AUROC on the MIMIC-CXR validation set as $t_\theta$.

We find a combination of random resized cropping and brightness/contrast adjustments (i.e. pixel distortion) to be the optimal pair of augmentations $t_\theta$ with an AUROC of 0.760 (Figure 2). Pairs of augmentations that include random resized cropping consistently outperform other compositions (AUROC improvements ranging from $0.023 - 0.092$). This in contrast to natural images, in which cropping performs well, but mostly in conjunction with either color jittering or Sobel filtering (Chen et al., 2020a). We further optimize the hyperparameters of $t_\theta$ and find that strong cropping ($scale = 0.2 - 0.5$) and large brightness/contrast distortions ($\lambda = 0.7$) are favored for single-branch augmentations, while weaker cropping ($scale = 0.3 - 0.9$) is favored for symmetrical dual-branch augmentations. A strategy without any augmentations yields surprisingly good results (0.668 AUROC) and serves as a baseline to compare other augmentations with. We report the Hamming Loss and Ranking Error for each of the augmentation pairs for MIMIC-CXR and CheXpert in the Appendix (Tables 8 and 10) and observe consistent trends.

Finally, we compare $t_\theta$ head-to-head with RandAugment and observed that RandAugment linear probing on MIMIC-CXR was effective (AUROC of 0.76) but not superior to the simpler $t_\theta$ strategy. We examined adding a third augmentation as well as several less common augmentations, but do not consider them for further experiments (Appendix C).

### 4.2. Comparison to Fully Supervised Networks

We compare linear probing performance on $t_\theta$ with fully supervised models trained from scratch [FS (S)] and with ImageNet pretrained weights [FS (IN)] on MIMIC-CXR (Table 1) and VinDr-CXR (Table 2). We observe that the linear probe on $t_\theta$ (or any pair of augmentations with resized cropping, see Appendix F) surpasses both fully supervised networks for MIMIC-CXR (by 0.050 and 0.018 AUROC) and VinDr-CXR (by 0.099 and 0.057 AUROC). Further stratifying the results, we show that the linear probes outperform the fully supervised approaches for almost all conditions, including the challenging minority class of rib fractures [RF] that has $< 5\%$ prevalence, by 0.089 and 0.055 AUROC (Table 1).

Table 1: Comparison of $t_\theta$ to fully supervised models on MIMIC-CXR. Numbers are AUROC. Abbreviations: FS: Fully supervised, S: trained from scratch, IN: ImageNet, AT: Atelectasis, CM: Cardiomegaly, RF: Rib fracture, PE: Pleural Effusion, PNA: Pneumonia, PTX: Pneumothorax.

| Strategy | AT | CM | Edema | RF | PE | PNA | PTX | No Finding | Macro AUROC |
|---|---|---|---|---|---|---|---|---|---|
| Linear probe | **0.750** | **0.769** | **0.848** | **0.619** | 0.845 | 0.649 | **0.779** | **0.819** | **0.760** |
| Fine tune | **0.799** | **0.806** | **0.880** | **0.678** | **0.892** | **0.725** | **0.841** | **0.856** | **0.810** |
| FS (S) | 0.705 | 0.720 | 0.797 | 0.529 | 0.837 | 0.617 | 0.689 | 0.786 | 0.710 |
| FS (IN) | 0.744 | 0.753 | 0.832 | 0.563 | 0.856 | 0.667 | 0.748 | 0.770 | 0.742 |

### 4.3. Zero-shot Generalization of Pretrained Representations

We evaluate zero-shot transfer of our $t_\theta$ representations to VinDr-CXR and CheXpert, which have differing disease distributions and dataset statistics. In zero-shot transfer to VinDr-CXR pathologies available in MIMIC-CXR, the $t_\theta$ representations achieve 0.805 AUROC, outperforming fully supervised VinDr-CXR networks by 0.094 and 0.002 AUROC when trained from scratch and ImageNet, respectively (Table 2). This was striking as the pretrained $t_\theta$ representations did not have any access to VinDr-CXR data or label distributions. However, such an effective zero-shot transfer was not the case with CheXpert. Here, the fully supervised CheXpert performance was higher than that of the $t_\theta$ representations (Table 3, *Eval on CheXpert*). We attribute this zero-shot discrepancy due to the substantially higher amount of labelled images in CheXpert than in VinDr-CXR (168,660 vs 18,000). Furthermore, the linear probes achieve better zero-shot generalization compared to fully supervised models, trained on MIMIC-CXR from scratch (Appendix F.2, Table 12).

### 4.4. Transferring Pretrained Representations to VinDr-CXR

Here, we linearly probe the $t_\theta$ representations pretrained on MIMIC-CXR using the VinDr-CXR dataset, which consists of seen and unseen pathologies. When evaluating the MIMIC-CXR $t_\theta$ pretrained classifiers on held out VinDr-CXR data, we observe 0.099 AUROC and 0.067 AUROC improvements over fully supervised models (same trends for CheXpert representations) in Table 2. This trend is consistent across almost all pathologies (we report additional results on different data splits in Appendix A, Table 7). Remarkably, the performance on Tuberculosis [TB], an unencountered disease with very low prevalence in the US, has 0.127 and 0.101 AUROC better performance than the supervised baselines (Table 2). This shows that linear probing of strong pretrained representations can generalize to out-of-distribution, unseen data and pathologies.

### 4.5. Generalization Capability by Fine-Tuning MIMIC-CXR $t_\theta$ on CheXpert

We fine-tune the pretrained MIMIC-CXR $t_\theta$ representations and MIMIC-CXR trained linear classifier on labelled CheXpert training data. We see that the classification AUROC increases on fine-tuning from 0.649 to 0.768 AUROC, even outperforming the fully supervised network trained from scratch (0.757 AUROC) (Table 3, *Eval on CheXpert*). We then evaluate this MIMIC-CXR $t_\theta$ pretrained and CheXpert fine-tuned model for its zero-shot transfer capabilities to evaluation data from MIMIC-CXR and VinDr-CXR data (Table

Table 2: Transferring $t_\theta$ to VinDr-CXR for seen (in-distribution) and unseen (out-of-distribution, OOD) conditions in MIMIC-CXR. Numbers are AUROC. Abbreviations: FS (S), FS (IN): Fully supervised from scratch and from ImageNet, respectively, CM: Cardiomegaly, PE: Pleural Effusion, PNA: Pneumonia, PF: Pulmonay Fibrosis, PT: Pleural Thicknening, LO: Lung Opacity, TB: Tuberculosis.

| | In-Distribution Pathologies | | | | Out-of-distribution Pathologies | | | | | | |
|---|---|---|---|---|---|---|---|---|---|---|---|
| Strategy | CM | PE | PNA | No finding | PF | PT | LO | Mass | TB | Macro AUROC | OOD AUROC |
| Zero-shot | 0.840 | 0.810 | 0.774 | 0.795 | NA | NA | NA | NA | NA | NA | NA |
| Linear probe | 0.909 | 0.822 | 0.785 | **0.880** | **0.720** | **0.712** | 0.651 | 0.648 | 0.776 | 0.767 | 0.701 |
| Fine tune | **0.937** | 0.824 | **0.790** | 0.869 | 0.719 | 0.707 | **0.660** | **0.651** | **0.802** | **0.773** | **0.708** |
| FS (S) | 0.796 | 0.643 | 0.591 | 0.813 | 0.631 | 0.665 | 0.622 | 0.598 | 0.649 | 0.668 | 0.633 |
| FS (IN) | 0.888 | **0.872** | 0.672 | 0.778 | 0.631 | 0.694 | 0.613 | 0.571 | 0.675 | 0.710 | 0.637 |

3). Upon fine-tuning these representations on CheXpert, we wish to assess the presence of catastrophic forgetting or poorer generalization to MIMIC-CXR through our zero-shot evaluations. However, we see that zero-shot evaluation on MIMIC-CXR continues showing high performance (0.763 AUROC), which still outperforms fully supervised models by 0.053 and 0.021 AUROC, indicating no evidence of catastrophic forgetting. In fact, the performance on MIMIC-CXR evaluation data is nearly identical (AUROC difference of 0.003) when fine-tuned on CheXpert or not. Similarly, this fine-tuned model generalizes well to VinDr-CXR with 0.810 AUROC, which is 0.142 and 0.091 AUROC higher than fully supervised baselines trained on VinDr-CXR (Table 3, *Eval on VinDr-CXR*). We believe that VinDR, being a relatively small dataset, benefits greatly from pretraining and fine-tuning as compared to fully supervised learning.

Table 3: MIMIC-CXR $t_\theta$ to CheXpert transfer on fine-tuning. Macro AUROC. Abbreviations: ZS: Zero-shot transfer, FT: Fine-tuning, FS: Fully Supervised, S: trained from scratch, IN: ImageNet, Chex: CheXpert, Mimic: MIMIC-CXR, VinDr: VinDr-CXR, Eval: Evaluation.

| Eval on CheXpert | | | | Eval on MIMIC-CXR | | | | Eval on VinDr-CXR | | | |
|---|---|---|---|---|---|---|---|---|---|---|---|
| ZS | FT (Chex) | FS (S) (Chex) | FS (IN) (Chex) | ZS | FT (Chex) | FS (S) (Mimic) | FS (IN) (Mimic) | ZS | FT (Chex) | FS (S) (VinDr) | FS (IN) (VinDr) |
| 0.649 | 0.768 | 0.757 | **0.789** | 0.760 | **0.763** | 0.710 | 0.742 | 0.765 | **0.810** | 0.668 | 0.719 |

## 4.6. Data-Efficiency in Fine-Tuning

We test the data-efficiency of our representations while fine-tuning, by varying the percentage of labelled data they are exposed to. We create stratified splits of the MIMIC-CXR training set, maintaining the label distribution, with 100%, 50%, 25%, 12.5%, 10% and 1% of the labelled data. All smaller subset splits are members of the larger split (i.e. all images in the 1% split are included in the 10% split, and so on). We fine-tune our MIMIC-CXR $t_\theta$ representations on each of the stratified splits from labelled MIMIC-CXR training data. We evaluate each fine-tuned network on held-out MIMIC-CXR validation data, and also assess zero-shot transfer on CheXpert and VinDr-CXR. We observe that fine-tuning, even with as little with 10% data, improves performance on all three datasets (Table 4), indicat-

ing that the representations are data-efficient. For CheXpert evaluation, we see that even 1% fine-tuning improves performance over zero-shot transfer by 0.03 AUROC. However, the fine-tuned performance still lags that of fully-supervised CheXpert, likely due to the scale of available training labels. Interestingly, 1% fine-tuning on VinDr-CXR reduces performance, while 10+% data improves performance. We hypothesize that this may be because the models overfits to the 1% data split and cannot generalize to distribution shifted manifold of VinDr-CXR, which has a varied label distribution (Table 6) than MIMIC-CXR.

Table 4: Fine-tuning data efficiency: Macro AUROC for fine-tuning MIMIC-CXR $t_\theta$ pretrained representations on three held out evaluation sets fine-tuned on stratified splits of MIMIC-CXR training data (in %). Results compared with zero-shot evaluations and fully supervised from scratch [FS (S)] or from ImageNet [FS (IN)] pretrained weights on their respective train sets.

| Eval Set | 1% | 10% | 12.5% | 25% | 50% | 100% | Zero-Shot | Linear Probe | FS (S) | FS (IN) |
|---|---|---|---|---|---|---|---|---|---|---|
| MIMIC-CXR | 0.783 | 0.792 | 0.797 | 0.800 | 0.805 | 0.810 | NA | 0.760 | 0.710 | 0.742 |
| CheXpert | 0.679 | 0.687 | 0.690 | 0.696 | 0.701 | 0.707 | 0.649 | 0.743 | 0.757 | 0.788 |
| VinDr-CXR | 0.740 | 0.773 | 0.786 | 0.792 | 0.805 | 0.803 | 0.765 | 0.767 | 0.668 | 0.710 |

## 5. Conclusion

In this work, we perform the first systematic exploration of augmentation strategies on the quality of self-supervised representations, across multiple datasets and mechanisms for chest x-rays. We find random resized cropping to be crucial, and adding random contrast and brightness adjustments yields powerful representations. The learned representations prove to be robust to out-of-distribution data, surpass the classification accuracy of fully supervised models for various disease labels, and even generalize to unseen conditions.

## Data and Code availability

We use publicly available, large scale chest X-ray datasets, MIMIC-CXR, CheXpert, and VinDr-CXR. Data collection details are given in Section 3.2 and data preprocessing steps are outlined in Appendix B. We open-source all our code for all our experiments and analyses in the paper on GitHub at https://github.com/StanfordMIMI/siaug.

## Acknowledgments

This work was supported by NIH grants R01 AR077604, R01 EB002524, R01 AR079431, K24 AR062068, and P41 EB027060; NIH contracts 75N92020C00008 and 75N92020C00021 and received computational support from Stability.AI and the Institute for Human-Centered AI at Stanford. RS received support from the Dutch Research Council, independent of this work. We would like to acknowledge Pierre Chambon for proofreading this manuscript and members of the Stanford MIMI group, Rubin Lab and Langlotz Lab for insightful discussions.

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

## Appendix A. Chest X-ray Dataset Distributions

Data distributions and splits used for model training and validation are outlined in Tables 5 and 6 below. We split both MIMIC-CXR and CheXpert datasets into training, validation and evaluation sets. Training sets are used for self-supervised pretraining, for which the labels are not utilized. We use the same training set to train linear probes, fine-tune pretrained representations, and supervised models, while using the associated labels. The validation set is used to track metrics in this phase and for hyperparameter tuning. The held-out evaluation sets are used for all evaluations including zero-shot transfer, linear probing, and fine-tuning. These evaluation splits are unseen during training and were never used to train representations, linear probes, or to fine-tune pretrained representations. For MIMIC-CXR, we use $n = 200,000$ images for training and validation, and $n = 37,962$ for evaluation. Similarly, for CheXpert, we use $n = 168,660$ images for training and validation, and $n = 22,367$ for evaluation.

Splits in the VinDr datasets were formed in two ways: balanced (i.e. stratified for the included conditions) and imbalanced. We term the VinDr-CXR dataset, *VinDr-Imbalanced*, as it consists of a large number "No Finding" labels ($\sim 60\%$ label prevalence). We create a separate subset of *VinDr-Balanced* by undersampling this majority class to $\sim 30\%$ label prevalence (all data splits in Table 6). We use *VinDr-Balanced* for all evaluations, linear-probing and fine-tuning.

We additionally report the performance of fine-tuning and fully supervised learning on the *VinDr-Imbalanced* set in the Appendix, Table 7. We report fully supervised performance

Table 5: Baseline characteristics of CheXpert and MIMIC-CXR.

| | MIMIC-CXR | | CheXpert | |
|---|---|---|---|---|
| Pathologies | Training & Validation | Evaluation | Training & Validation | Evaluation |
| Atelectasis | 48,833 (24%) | 9461 (25%) | 51,892 (30%) | 7691 (34%) |
| Cardiomegaly | 44,206 (22%) | 8404 (22%) | 26,989 (16%) | 3103 (14%) |
| Edema | 35,440 (18%) | 6718 (18%) | 54,755 (33%) | 6738 (30%) |
| Pleural Effusion | 51,836 (26%) | 10,306 (27%) | 78,258 (46%) | 8219 (37%) |
| Pneumonia | 29,969 (15%) | 5704 (15%) | 18,235 (11%) | 2421 (11%) |
| Pneumothorax | 10,294 (5%) | 1929 (5%) | 18,674 (11%) | 1727 (8%) |
| Rib Fracture | 4444 (2%) | 825 (2%) | 6914 (4%) | 1021 (5%) |
| No Finding | 67,239 (34%) | 12,647 (33%) | 14,430 (9%) | 2544 (11%) |

on *VinDr-Imbalanced* in the main paper for the following reasons: (a) fully supervised performance on VinDR-Imbalanced is higher than that on VinDR-Balanced (Table 7). This is expected as VinDr-Imbalanced contains more data, i.e. of the majority class, than VinDR-Balanced. (b) By showing that we outperform the best possible fully supervised performance (i.e. by using VinDR-Imbalanced) in the main paper, we also implicitly outperform fully supervised performance on VinDR-Balanced as given here in the Appendix, Table 7.

Table 6: VinDr-CXR splits distribution. **Bold** refers to unseen concepts.

| | VinDR-Balanced | | VinDR-Imbalanced | |
|---|---|---|---|---|
| Pathologies | Training | Evaluation | Training | Evaluation |
| **Pulmonary fibrosis** | 1017 (11%) | 217 (11%) | 1017 (6%) | 217 (6%) |
| Cardiomegaly | 1817 (20%) | 309 (16%) | 1817 (11%) | 309 (9%) |
| **Pleural thickening** | 882 (10%) | 169 (9%) | 882 (5%) | 169 (5%) |
| **Lung Opacity** | 547 (6%) | 84 (4%) | 547 (3%) | 84 (2%) |
| Pleural effusion | 634 (7%) | 111 (6%) | 634 (4%) | 111 (3%) |
| Pneumonia | 471 (5%) | 246 (12%) | 471 (3%) | 246 (7%) |
| **Tuberculosis** | 482 (5%) | 164 (8%) | 482 (3%) | 164 (5%) |
| **Nodule/Mass** | 409 (4%) | 176 (9%) | 409 (3%) | 176 (5%) |
| No finding | 3000 (32%) | 500 (25%) | 10601 (63%) | 2051 (58%) |

Table 7: Fully supervised performance and Fine-tuning from MIMIC-CXR $t_\theta$ representations performance on *VinDr-Balanced* and *VinDr-Imbalanced* datasplits. Numbers are AUROC. Abbreviations: FS (S), FS (IN): Fully supervised from scratch and from ImageNet, respectively, CM: Cardiomegaly, PE: Pleural Effusion, PNA: Pneumonia, PF: Pulmonay Fibrosis, PT: Pleural Thicknening, LO: Lung Opacity, TB: Tuberculosis.

| | | In-Distribution Pathologies | | | | Out-of-dsitribution Pathologies | | | | | | |
|---|---|---|---|---|---|---|---|---|---|---|---|---|
| Dataset | Strategy | CM | PE | PNA | No Finding | PF | PT | LO | Mass | TB | Macro AUROC | OOD AUROC |
| *VinDr-Balanced* | FS (S) | 0.796 | 0.644 | 0.610 | 0.732 | 0.601 | 0.633 | 0.616 | 0.553 | 0.648 | 0.648 | 0.610 |
| | FS (IN) | 0.871 | 0.731 | 0.659 | 0.801 | 0.643 | 0.651 | 0.565 | 0.615 | 0.680 | 0.690 | 0.631 |
| | FT | **0.937** | **0.824** | **0.790** | **0.869** | **0.719** | 0.707 | 0.660 | **0.651** | **0.802** | **0.773** | **0.708** |
| *VinDr-Imbalanced* | FS (S) | 0.796 | 0.643 | 0.591 | 0.813 | 0.631 | 0.665 | 0.622 | 0.598 | 0.649 | 0.668 | 0.633 |
| | FS (IN) | 0.888 | **0.872** | 0.672 | 0.778 | 0.631 | 0.694 | 0.613 | 0.571 | 0.675 | 0.710 | 0.637 |
| | FT | **0.917** | 0.810 | **0.777** | **0.868** | **0.714** | **0.708** | **0.661** | **0.649** | **0.797** | **0.767** | **0.706** |

## Appendix B. Data Pre-Processing

Data was acquired in DICOM format for MIMIC-CXR and VinDr-CXR, while CheXpert images had been obtained as pre-processed images in JPEG format. Data was pre-processed on the basis of the DICOM headers. Images were corrected for photometric interpretation, windowed according to their respective window center and width, and scaled with an intercept and slope, if applicable. All images were resized to 224x224 pixels.

## Appendix C. Augmentations

The augmentations referred to in the main text were implemented using the Kornia library (Riba et al., 2020) for PyTorch. We explored the addition of a third augmentation from a broader set of augmentations to $t_\theta$ strategy and found that it did not yield higher performance. As a result, we did not explore further addition of augmentations. Apart from random resized cropping, rotation, cutout, contrast/brightness adjustments, Gaussian noise, Gaussian blur, and Sobel filtering, we explored an additional set of less commonly used augmentations, including thin plate spline transforms (Riba et al., 2020), motion blur, jigsaw puzzles (Noroozi and Favaro, 2016), and plasma fractals (Nicolaou et al., 2022). These augmentations were evaluated on an individual basis as an add-on to the augmentations of $t_\theta$. None of these augmentations surpassed the performance of $t_\theta$ while linear probing MIMIC-CXR. We note that more complicated strategies based on our findings (e.g., tuning an augmentation policy) could potentially lead to further improvements, but would sacrifice the simplicity of the current approach. We hope to explore this in future work.

### C.1. Implementation Details

We use `RandomResizedCrop` to construct crops with random scale of $0.2 - 1.0$ for pair-wise evaluation and $0.3 - 0.9$ for $t_\theta$, and the default parameters for aspect ratio ($3/4 - 4/3$). Brightness/contrast adjustment is implemented using `ColorJitter` with brightness and contrast arguments ($\lambda$) set to 0.5 for pair-wise evaluations and $t_\theta$. A kernel size of 23 with a sigma of $0.1 - 2.0$ was used for Gaussian blur, whereas Gaussian noise had $\mu$ set to 0 and $\sigma$ uniformly sampled from $0.01 - 0.03$. Cutout was implemented by `RandomErasing` with the default parameters (scale range $0.02 - 0.33$, ratio range $0.3 - 3.3$). All other augmentations were implemented using the default parameters as supplied by the Kornia library.

### C.2. RandAugment

The RandAugment (Cubuk et al., 2020) strategy was applied using all augmentations mentioned in Section C.1 with the number of of augmentations ($n$) set to 3, and a magnitude defined by the same hyperparameters as described above.

## Appendix D. Additional Training Details

Representations pretrained for optimal augmentation selection were trained for 50 epochs whose training duration ranged from approximately 6 to 12 hours. The corresponding linear probes were trained for 40 epochs. Checkpoints from the final epochs were used for evaluation. The $t_\theta$ set of augmentations was retrained for 100 epochs, and linear probes were

trained for 90 epochs. Linear probes, fine-tuned models, and fully supervised models were trained free of augmentations to investigate the effectiveness of the pretrained embeddings. Fine-tuned and supervised models were trained for 90 (MIMIC-CXR and CheXpert) and 150 (VinDr-CXR) epochs.

## Appendix E. Multi-Label Metrics

Evaluating multi-label classification performance is more nuanced than typical multi-class classification scenarios. Common metrics such as accuracy and AUROC might overestimate or underestimate classifier capability. As a result, we report on three multi-label metrics including AUROC, that cover three different aspects of the classifier predictions. The **Hamming loss** is an example-based metric (Tsoumakas et al., 2010) and computes the fraction of misclassified labels across each sample and across each label. The lower the Hamming loss, the better. It is mathematically defined as $H = \frac{1}{NK} \sum_{i=i}^{n} \sum_{j=1}^{K} [p_{ij} \neq y_{ij}]$ where $p_{ij}$ is the prediction, $y_{ij}$ is the label, $K$ is the number of classes and $N$ is the number of samples. The **Ranking Error** (Tsoumakas et al., 2010) is a ranking type of metric that computes the number of times the irrelevant labels (i.e., low probability labels) are ranked higher than relevant labels. The lower the Ranking error, the better.

## Appendix F. Additional Results

### F.1. Pairwise Augmentations

We report the performance of pairwise augmentations with MIMIC-CXR pretraining followed by linear probing using three different metrics: Macro AUROC, Hamming Loss and Ranking Error in Table 8. We also report the class-wise AUROC scores for each of these pairs of augmentations on MIMIC-CXR pretraining and linear probing in Table 9. Similarly, performance of pairwise augmentations with CheXpert pretraining and linear probing are reported in Tables 10 and 11.

### F.2. Zero-shot Generalization Comparison

We show in Table 12 that linear probes trained on $t_\theta$ representations that were pretrained on MIMIC-CXR achieve better zero-shot generalization to other datasets (CheXpert and VinDr-CXR), compared to fully supervised models trained from scratch on MIMIC-CXR as well as those initialized with ImageNet pretrained weights. This strongly suggests that augmentation-based pretraining leads to more generalizable models.

### F.3. Generalization and Fine-tuning Results for Pairwise Augmentations

We compare various pairwise augmentations in Section 4.1 to determine the optimal $t_\theta$ representations by evaluating the downstream classification performance of linear probes, trained on top of the representations. Here, we further compare the different pairs of augmentations on their generalization capabilities and fine-tuning performance. We choose a subset of these pairs (three containing Crop & Resize and three without), namely: (a)

Distort-Sobel, (b) Rotate-Distort, (c) Rotate-Sobel, (d) Crop & Resize-Noise, (e) Sobel-Crop & Resize, and (f) Crop & Resize-Distort for further evaluation. We report the fine-tuning performance in Tables 13 and 14 and the generalization capabilties and performance of their linear probes in Table 15 and Table 16 respectively.

We see that augmentation pairs with Crop & Resize outperform augmentations without Crop & Resize consistently, following the same trend in Section 4.1. This strongly suggests that Crop & Resize must be one of the augmentations in the pair. Indeed, while in majority of the cases, augmentations Crop & Resize and Distort (which constitute the best pair of augmentations in our experiments) outperform other pairs, in some cases, the augmentation pair of Sobel and Crop & Resize shows higher performance. However, we still believe that the combination of Crop & Resize and Distort may be superior overall. This is because Distort can be applied in various levels, giving a degree of flexibility and control to the practitioner in selecting those levels via hyperparameter setting unlike Sobel which has limited flexibility.

### F.4. Generalization to Siamese Representation Learning Strategies

Here, we evaluate whether the augmentation strategy $t_\theta$ generalizes to other Siamese representation learning strategies. We consider three commonly used frameworks: SimCLR (Chen et al., 2020a), DINO (Caron et al., 2021), and MoCo (Chen et al., 2020b). Our findings show that $t_\theta$ generalizes well to a variety of pretraining strategies, including those that rely on negative pairs, such as SimCLR (Table 17). All strategies were trained with the default settings with a $ResNet50$ backbone for 100 epochs with a batch size of 256. The DINO framework was trained as outlined in (Caron et al., 2021) with six local crops, and in a setting with global crops only.

### F.5. Linear Probing results on CheXpert

We expand Table 3 in Section 4.5, to include Linear Probing results with $t_\theta$ representations on CheXpert data in Table 18. We see that Linear probing results in some degree of catastrophic forgetting while fine-tuning does not.

Table 8: Pairwise Augmentations Performance with MIMIC-CXR pretraining and linear-probing

| Augmentation 1 | Augmentation 2 | Macro AUROC ↑ | Hamming Loss ↓ | Ranking Error ↓ |
|---|---|---|---|---|
| Fully Supervised (Scratch) | | 0.71 | 0.186 | 0.291 |
| Fully Supervised (ImageNet) | | 0.742 | 0.174 | 0.226 |
| Blur | Cutout | 0.676 | 0.181 | 0.224 |
| Blur | Identity | 0.671 | 0.191 | 0.237 |
| Blur | Jitter | 0.697 | 0.181 | 0.216 |
| Blur | Noise | 0.675 | 0.192 | 0.239 |
| Blur | Rotate | 0.676 | 0.192 | 0.233 |
| Blur | Crop & Resize | 0.749 | 0.166 | 0.181 |
| Blur | Sobel | 0.686 | 0.177 | 0.216 |
| Cutout | Blur | 0.68 | 0.182 | 0.224 |
| Cutout | Identity | 0.671 | 0.187 | 0.231 |
| Cutout | Distort | 0.686 | 0.186 | 0.223 |
| Cutout | Noise | 0.675 | 0.186 | 0.23 |
| Cutout | Rotate | 0.703 | 0.182 | 0.218 |
| Cutout | Crop & Resize | 0.747 | 0.171 | 0.189 |
| Cutout | Sobel | 0.69 | 0.176 | 0.214 |
| Identity | Identity | 0.668 | 0.187 | 0.234 |
| Distort | Blur | 0.695 | 0.181 | 0.217 |
| Distort | Cutout | 0.692 | 0.183 | 0.218 |
| Distort | Identity | 0.694 | 0.18 | 0.217 |
| Distort | Noise | 0.696 | 0.18 | 0.216 |
| Distort | Rotate | 0.709 | 0.184 | 0.215 |
| Distort | Crop & Resize | **0.76** | **0.163** | **0.175** |
| Distort | Sobel | 0.708 | 0.174 | 0.205 |
| Noise | Blur | 0.674 | 0.189 | 0.233 |
| Noise | Cutout | 0.676 | 0.185 | 0.232 |
| Noise | Identity | 0.671 | 0.192 | 0.237 |
| Noise | Distort | 0.696 | 0.18 | 0.215 |
| Noise | Rotate | 0.673 | 0.192 | 0.24 |
| Noise | Crop & Resize | 0.748 | 0.166 | 0.184 |
| Noise | Sobel | 0.686 | 0.175 | 0.215 |
| Rotate | Blur | 0.681 | 0.19 | 0.23 |
| Rotate | Cutout | 0.698 | 0.183 | 0.22 |
| Rotate | Identity | 0.668 | 0.196 | 0.245 |
| Rotate | Distort | 0.714 | 0.183 | 0.213 |
| Rotate | Noise | 0.674 | 0.191 | 0.237 |
| Rotate | Crop & Resize | 0.745 | 0.168 | 0.186 |
| Rotate | Sobel | 0.705 | 0.179 | 0.21 |
| Crop & Resize | Blur | 0.751 | 0.165 | 0.18 |
| Crop & Resize | Cutout | 0.747 | 0.173 | 0.19 |
| Crop & Resize | Identity | 0.745 | 0.169 | 0.187 |
| Crop & Resize | Distort | **0.761** | **0.16** | **0.174** |
| Crop & Resize | Noise | 0.754 | 0.165 | 0.179 |
| Crop & Resize | Rotate | 0.748 | 0.168 | 0.184 |
| Crop & Resize | Sobel | 0.738 | 0.167 | 0.187 |
| Sobel | Blur | 0.686 | 0.178 | 0.218 |
| Sobel | Cutout | 0.692 | 0.176 | 0.213 |
| Sobel | Identity | 0.685 | 0.176 | 0.217 |
| Sobel | Distort | 0.701 | 0.179 | 0.213 |
| Sobel | Noise | 0.684 | 0.176 | 0.217 |
| Sobel | Rotate | 0.701 | 0.182 | 0.215 |
| Sobel | Crop & Resize | 0.755 | 0.161 | 0.174 |

Table 9: Class-wise AUROC for pairwise augmentations with MIMIC-CXR pretraining and linear-probing. Abbreviations: AT: Atelectasis, CM: Cardiomegaly, RF: Rib fracture, PE: Pleural Effusion, PNA: Pneumonia, PTX: Pneumothorax.

| Augmentation 1 | Augmentation 2 | AT | CM | Edema | RF | PE | PNA | PTX | No Finding |
|---|---|---|---|---|---|---|---|---|---|
| Fully Supervised (Scratch) | | 0.705 | 0.72 | 0.797 | 0.529 | 0.837 | 0.617 | 0.689 | 0.786 |
| Fully Supervised (ImageNet) | | 0.744 | 0.753 | 0.832 | 0.563 | 0.856 | 0.667 | 0.748 | 0.77 |
| Blur | Cutout | 0.675 | 0.694 | 0.753 | 0.544 | 0.733 | 0.571 | 0.695 | 0.742 |
| Blur | Identity | 0.662 | 0.684 | 0.763 | 0.552 | 0.731 | 0.567 | 0.678 | 0.728 |
| Blur | Distort | 0.685 | 0.706 | 0.787 | 0.569 | 0.763 | 0.581 | 0.719 | 0.764 |
| Blur | Noise | 0.666 | 0.686 | 0.766 | 0.547 | 0.733 | 0.579 | 0.687 | 0.736 |
| Blur | Rotate | 0.671 | 0.68 | 0.765 | 0.56 | 0.738 | 0.576 | 0.68 | 0.74 |
| Blur | Crop & Resize | 0.741 | 0.766 | 0.838 | 0.599 | 0.832 | 0.642 | 0.763 | 0.812 |
| Blur | Sobel | 0.678 | 0.7 | 0.768 | 0.559 | 0.739 | 0.586 | 0.704 | 0.752 |
| Cutout | Blur | 0.675 | 0.697 | 0.761 | 0.546 | 0.74 | 0.575 | 0.699 | 0.746 |
| Cutout | Identity | 0.665 | 0.686 | 0.746 | 0.557 | 0.722 | 0.57 | 0.69 | 0.73 |
| Cutout | Distort | 0.675 | 0.701 | 0.769 | 0.56 | 0.743 | 0.583 | 0.704 | 0.754 |
| Cutout | Noise | 0.67 | 0.688 | 0.754 | 0.554 | 0.727 | 0.578 | 0.693 | 0.736 |
| Cutout | Rotate | 0.684 | 0.713 | 0.793 | 0.573 | 0.771 | 0.595 | 0.728 | 0.768 |
| Cutout | Crop & Resize | 0.732 | 0.749 | 0.833 | 0.614 | 0.828 | 0.648 | 0.768 | 0.801 |
| Cutout | Sobel | 0.682 | 0.704 | 0.774 | 0.56 | 0.743 | 0.592 | 0.705 | 0.757 |
| Identity | Identity | 0.662 | 0.681 | 0.759 | 0.542 | 0.724 | 0.569 | 0.677 | 0.73 |
| Distort | Blur | 0.681 | 0.706 | 0.788 | 0.567 | 0.759 | 0.585 | 0.707 | 0.766 |
| Distort | Cutout | 0.677 | 0.703 | 0.779 | 0.563 | 0.748 | 0.589 | 0.714 | 0.762 |
| Distort | Identity | 0.683 | 0.703 | 0.783 | 0.563 | 0.758 | 0.587 | 0.711 | 0.766 |
| Distort | Noise | 0.683 | 0.707 | 0.788 | 0.566 | 0.762 | 0.591 | 0.705 | 0.767 |
| Distort | Rotate | 0.698 | 0.718 | 0.803 | 0.567 | 0.778 | 0.611 | 0.721 | 0.779 |
| Distort | Crop & Resize | 0.75 | 0.769 | 0.848 | 0.619 | 0.845 | 0.649 | 0.779 | 0.819 |
| Distort | Sobel | 0.695 | 0.714 | 0.797 | 0.582 | 0.77 | 0.602 | 0.726 | 0.775 |
| Noise | Blur | 0.668 | 0.685 | 0.764 | 0.558 | 0.733 | 0.565 | 0.683 | 0.734 |
| Noise | Cutout | 0.673 | 0.687 | 0.755 | 0.565 | 0.728 | 0.577 | 0.691 | 0.735 |
| Noise | Identity | 0.665 | 0.684 | 0.762 | 0.54 | 0.734 | 0.574 | 0.673 | 0.735 |
| Noise | Distort | 0.681 | 0.707 | 0.788 | 0.565 | 0.761 | 0.592 | 0.707 | 0.765 |
| Noise | Rotate | 0.667 | 0.68 | 0.764 | 0.563 | 0.731 | 0.573 | 0.669 | 0.734 |
| Noise | Crop & Resize | 0.738 | 0.765 | 0.839 | 0.615 | 0.832 | 0.634 | 0.757 | 0.807 |
| Noise | Sobel | 0.682 | 0.7 | 0.765 | 0.56 | 0.739 | 0.589 | 0.703 | 0.754 |
| Rotate | Blur | 0.672 | 0.684 | 0.766 | 0.562 | 0.745 | 0.583 | 0.689 | 0.745 |
| Rotate | Cutout | 0.682 | 0.705 | 0.785 | 0.564 | 0.769 | 0.591 | 0.729 | 0.765 |
| Rotate | Identity | 0.662 | 0.675 | 0.75 | 0.555 | 0.722 | 0.57 | 0.684 | 0.726 |
| Rotate | Distort | 0.7 | 0.727 | 0.809 | 0.565 | 0.786 | 0.617 | 0.731 | 0.779 |
| Rotate | Noise | 0.668 | 0.682 | 0.764 | 0.568 | 0.73 | 0.573 | 0.675 | 0.73 |
| Rotate | Crop & Resize | 0.731 | 0.756 | 0.84 | 0.6 | 0.824 | 0.64 | 0.765 | 0.803 |
| Rotate | Sobel | 0.689 | 0.711 | 0.794 | 0.578 | 0.773 | 0.598 | 0.726 | 0.772 |
| Crop & Resize | Blur | 0.741 | 0.767 | 0.841 | 0.604 | 0.833 | 0.644 | 0.767 | 0.809 |
| Crop & Resize | Cutout | 0.734 | 0.749 | 0.835 | 0.612 | 0.828 | 0.649 | 0.768 | 0.803 |
| Crop & Resize | Identity | 0.732 | 0.76 | 0.835 | 0.609 | 0.824 | 0.636 | 0.76 | 0.805 |
| Crop & Resize | Distort | 0.753 | 0.771 | 0.849 | 0.608 | 0.844 | 0.655 | 0.786 | 0.821 |
| Crop & Resize | Noise | 0.743 | 0.766 | 0.843 | 0.617 | 0.834 | 0.644 | 0.768 | 0.813 |
| Crop & Resize | Rotate | 0.734 | 0.762 | 0.843 | 0.601 | 0.823 | 0.643 | 0.771 | 0.808 |
| Crop & Resize | Sobel | 0.727 | 0.754 | 0.832 | 0.589 | 0.809 | 0.631 | 0.753 | 0.806 |
| Sobel | Blur | 0.678 | 0.699 | 0.764 | 0.562 | 0.74 | 0.585 | 0.708 | 0.753 |
| Sobel | Cutout | 0.684 | 0.701 | 0.778 | 0.563 | 0.749 | 0.593 | 0.71 | 0.759 |
| Sobel | Identity | 0.681 | 0.701 | 0.766 | 0.557 | 0.737 | 0.585 | 0.705 | 0.751 |
| Sobel | Distort | 0.685 | 0.707 | 0.784 | 0.584 | 0.766 | 0.595 | 0.721 | 0.771 |
| Sobel | Noise | 0.681 | 0.699 | 0.765 | 0.556 | 0.736 | 0.585 | 0.702 | 0.748 |
| Sobel | Rotate | 0.683 | 0.708 | 0.789 | 0.578 | 0.767 | 0.595 | 0.724 | 0.766 |
| Sobel | Crop & Resize | 0.745 | 0.773 | 0.849 | 0.603 | 0.829 | 0.656 | 0.763 | 0.823 |

Table 10: Pairwise Augmentations Performance with CheXpert pretraining and linear-probing

| Augmentation 1 | Augmentation 2 | Macro AUROC ↑ | Hamming Loss↓ | Ranking Error ↓ |
|---|---|---|---|---|
| Fully Supervised (Scratch) | | 0.757 | 0.16 | 0.165 |
| Fully Supervised (ImageNet) | | 0.788 | 0.153 | 0.145 |
| Distort | Blur | 0.64 | 0.183 | 0.244 |
| Distort | Cutout | 0.604 | 0.187 | 0.258 |
| Distort | Noise | 0.661 | 0.181 | 0.228 |
| Distort | Rotate | 0.668 | 0.179 | 0.227 |
| Distort | Crop & Resize | **0.736** | **0.167** | **0.189** |
| Distort | Sobel | 0.666 | 0.179 | 0.233 |
| Noise | Blur | 0.66 | 0.181 | 0.229 |
| Noise | Cutout | 0.653 | 0.182 | 0.233 |
| Noise | Noise | 0.676 | 0.179 | 0.219 |
| Noise | Rotate | 0.663 | 0.181 | 0.227 |
| Noise | Crop & Resize | 0.691 | 0.176 | 0.214 |
| Noise | Sobel | 0.594 | 0.186 | 0.263 |
| Crop & Resize | Blur | 0.713 | 0.172 | 0.203 |
| Crop & Resize | Cutout | 0.73 | 0.169 | 0.193 |
| Crop & Resize | Noise | 0.736 | 0.166 | 0.186 |
| Crop & Resize | Rotate | 0.688 | 0.176 | 0.215 |
| Crop & Resize | Crop & Resize | 0.708 | 0.174 | 0.206 |
| Crop & Resize | Sobel | 0.725 | 0.17 | 0.195 |
| Sobel | Blur | 0.631 | 0.184 | 0.248 |
| Sobel | Cutout | 0.616 | 0.187 | 0.261 |
| Sobel | Noise | 0.651 | 0.182 | 0.238 |
| Sobel | Rotate | 0.517 | 0.213 | 0.432 |
| Sobel | Crop & Resize | 0.638 | 0.183 | 0.245 |
| Sobel | Sobel | 0.712 | 0.171 | 0.201 |

Table 11: Class-wise AUROC for pairwise augmentations with CheXpert pretraining and linear-probing. Abbreviations: AT: Atelectasis, CM: Cardiomegaly, RF: Rib fracture, PE: Pleural Effusion, PNA: Pneumonia, PTX: Pneumothorax.

| Augmentation 1 | Augmentation 2 | AT | CM | Edema | RF | PE | PNA | PTX | No Finding |
|---|---|---|---|---|---|---|---|---|---|
| Fully Supervised (Scratch) | | 0.685 | 0.792 | 0.787 | 0.679 | 0.84 | 0.703 | 0.73 | 0.839 |
| Fully Supervised (ImageNet) | | 0.703 | 0.824 | 0.816 | 0.72 | 0.862 | 0.743 | 0.783 | 0.857 |
| Distort | Blur | 0.604 | 0.622 | 0.675 | 0.623 | 0.679 | 0.6 | 0.626 | 0.69 |
| Distort | Cutout | 0.569 | 0.579 | 0.62 | 0.582 | 0.643 | 0.584 | 0.614 | 0.644 |
| Distort | Noise | 0.621 | 0.667 | 0.691 | 0.627 | 0.697 | 0.602 | 0.637 | 0.742 |
| Distort | Rotate | 0.626 | 0.649 | 0.697 | 0.646 | 0.704 | 0.628 | 0.659 | 0.733 |
| Distort | Crop & Resize | 0.668 | 0.759 | 0.772 | 0.693 | 0.786 | 0.68 | 0.716 | 0.818 |
| Distort | Sobel | 0.612 | 0.66 | 0.714 | 0.634 | 0.708 | 0.615 | 0.654 | 0.736 |
| Noise | Blur | 0.621 | 0.672 | 0.686 | 0.627 | 0.701 | 0.596 | 0.634 | 0.74 |
| Noise | Cutout | 0.618 | 0.653 | 0.685 | 0.623 | 0.691 | 0.592 | 0.63 | 0.731 |
| Noise | Distort | 0.625 | 0.687 | 0.712 | 0.639 | 0.713 | 0.623 | 0.647 | 0.762 |
| Noise | Rotate | 0.625 | 0.669 | 0.69 | 0.636 | 0.705 | 0.597 | 0.633 | 0.749 |
| Noise | Crop & Resize | 0.644 | 0.71 | 0.72 | 0.655 | 0.735 | 0.633 | 0.656 | 0.776 |
| Noise | Sobel | 0.574 | 0.563 | 0.605 | 0.583 | 0.631 | 0.557 | 0.594 | 0.647 |
| Crop & Resize | Blur | 0.651 | 0.723 | 0.752 | 0.675 | 0.757 | 0.663 | 0.688 | 0.796 |
| Crop & Resize | Cutout | 0.65 | 0.722 | 0.769 | 0.704 | 0.773 | 0.689 | 0.727 | 0.803 |
| Crop & Resize | Distort | 0.666 | 0.76 | 0.772 | 0.682 | 0.79 | 0.682 | 0.715 | 0.82 |
| Crop & Resize | Noise | 0.64 | 0.705 | 0.719 | 0.654 | 0.731 | 0.632 | 0.655 | 0.772 |
| Crop & Resize | Rotate | 0.641 | 0.709 | 0.746 | 0.684 | 0.748 | 0.664 | 0.693 | 0.782 |
| Crop & Resize | Sobel | 0.655 | 0.74 | 0.764 | 0.684 | 0.764 | 0.677 | 0.715 | 0.799 |
| Sobel | Blur | 0.587 | 0.589 | 0.66 | 0.616 | 0.668 | 0.594 | 0.646 | 0.689 |
| Sobel | Cutout | 0.579 | 0.59 | 0.628 | 0.6 | 0.649 | 0.576 | 0.652 | 0.655 |
| Sobel | Distort | 0.605 | 0.642 | 0.685 | 0.613 | 0.689 | 0.602 | 0.652 | 0.719 |
| Sobel | Noise | 0.517 | 0.518 | 0.512 | 0.5 | 0.57 | 0.506 | 0.515 | 0.501 |
| Sobel | Rotate | 0.596 | 0.607 | 0.663 | 0.626 | 0.68 | 0.592 | 0.638 | 0.703 |
| Sobel | Crop & Resize | 0.651 | 0.729 | 0.75 | 0.662 | 0.761 | 0.653 | 0.697 | 0.796 |

Table 12: Comparison of zero-shot generalization capabilties of linear probes on MIMIC-CXR pretrained $t_\theta$ representations (Linear Probe) to those of fully supervised models trained on MIMIC-CXR from scratch [FS (S)] or from ImageNet pretrained weights [FS (IN)]

| | | Evaluation Data | | |
|---|---|---|---|---|
| Strategy | (Pre)Training Data | MIMIC-CXR | CheXpert | VinDr-CXR |
| Linear Probe | MIMIC-CXR | **0.760** | **0.649** | **0.765** |
| FS (S) | MIMIC-CXR | 0.710 | 0.612 | 0.668 |
| FS (IN) | MIMIC-CXR | 0.742 | 0.630 | 0.710 |

Table 13: Fine-tuning performance of augmentation pairs on MIMIC-CXR. Numbers are Macro AUROCs. All models are pretrained and fine-tuned on MIMIC-CXR data.

| Pretrained on MIMIC-CXR Fine-Tuned on MIMIC-CXR | | Evaluation Data | | |
|---|---|---|---|---|
| Augmentation 1 | Augmentation 2 | MIMIC-CXR | CheXpert | VinDr-CXR |
| Distort | Sobel | 0.762 | 0.660 | 0.762 |
| Rotate | Distort | 0.765 | 0.661 | 0.758 |
| Rotate | Sobel | 0.738 | 0.646 | 0.746 |
| Crop & Resize | Noise | 0.791 | 0.697 | 0.791 |
| Sobel | Crop & Resize | 0.801 | 0.687 | **0.807** |
| Crop & Resize | Distort | **0.810** | **0.707** | 0.803 |

Table 14: Fine-tuning performance of augmentation pairs on VinDr-CXR. Models are pretrained on MIMIC-CXR and fine-tuned on VinDr-CXR. Numbers are AUROC. Abbreviations: Aug: Augmentation, CM: Cardiomegaly, PE: Pleural Effusion, PNA: Pneumonia, PF: Pulmonay Fibrosis, PT: Pleural Thicknening, LO: Lung Opacity, TB: Tuberculosis.

| Pretrained on MIMIC-CXR Fine-Tuned on VinDr-CXR | | Evaluation Data: VinDr-CXR | | | | | | | | | |
|---|---|---|---|---|---|---|---|---|---|---|---|
| Aug 1 | Aug 2 | Macro AUROC | CM | PE | PNA | No Finding | PF | PT | LO | Mass | TB |
| Distort | Sobel | 0.684 | 0.828 | 0.670 | 0.618 | 0.780 | 0.649 | 0.664 | 0.631 | 0.597 | 0.723 |
| Rotate | Distort | 0.698 | 0.853 | 0.687 | 0.705 | 0.801 | 0.638 | 0.630 | 0.651 | 0.611 | 0.711 |
| Rotate | Sobel | 0.670 | 0.824 | 0.671 | 0.630 | 0.764 | 0.631 | 0.649 | 0.606 | 0.579 | 0.678 |
| Crop & Resize | Noise | 0.766 | 0.920 | 0.830 | 0.773 | 0.883 | 0.706 | 0.708 | 0.676 | 0.643 | 0.759 |
| Sobel | Crop & Resize | **0.790** | **0.939** | **0.837** | **0.802** | **0.889** | **0.736** | **0.752** | **0.701** | 0.649 | **0.803** |
| Crop & Resize | Distort | 0.773 | 0.937 | 0.824 | 0.790 | 0.869 | 0.719 | 0.707 | 0.660 | **0.651** | 0.802 |

Table 15: Generalization performance of linear probes on MIMIC-CXR for different augmentation pairs. Numbers are Macro AUROCs. All models are pretrained and linearly probed on MIMIC-CXR data.

| Pretrained on MIMIC-CXR Linear Probes on MIMIC-CXR | | Evaluation Data | | |
|---|---|---|---|---|
| Augmentation 1 | Augmentation 2 | MIMIC-CXR | CheXpert | VinDr-CXR |
| Distort | Sobel | 0.708 | 0.637 | 0.679 |
| Rotate | Distort | 0.714 | 0.635 | 0.697 |
| Rotate | Sobel | 0.705 | 0.613 | 0.655 |
| Crop & Resize | Noise | 0.754 | **0.656** | 0.741 |
| Sobel | Crop & Resize | 0.755 | 0.653 | 0.760 |
| Crop & Resize | Distort | **0.761** | 0.652 | **0.765** |

Table 16: Generalization performance of linear probes on VinDr-CXR for different augmentation pairs. Models are pretrained on MIMIC-CXR and linear probes are trained on VinDr-CXR. Numbers are AUROC. Abbreviations: Aug: Augmentation, CM: Cardiomegaly, PE: Pleural Effusion, PNA: Pneumonia, PF: Pulmonay Fibrosis, PT: Pleural Thicknening, LO: Lung Opacity, TB: Tuberculosis.

| Pretrained on MIMIC-CXR Linear Probes on VinDr-CXR | | Evaluation Data: VinDr-CXR | | | | | | | | | |
|---|---|---|---|---|---|---|---|---|---|---|---|
| Aug 1 | Aug 2 | Macro AUROC | CM | PE | PNA | No Finding | PF | PT | LO | Mass | TB |
| Distort | Sobel | 0.663 | 0.836 | 0.646 | 0.594 | 0.769 | 0.617 | 0.635 | 0.613 | 0.592 | 0.666 |
| Rotate | Distort | 0.675 | 0.853 | 0.649 | 0.673 | 0.775 | 0.631 | 0.613 | 0.626 | 0.578 | 0.675 |
| Rotate | Sobel | 0.638 | 0.803 | 0.667 | 0.595 | 0.690 | 0.586 | 0.600 | 0.596 | 0.569 | 0.635 |
| Crop & Resize | Noise | 0.748 | 0.897 | 0.801 | 0.756 | 0.882 | 0.662 | 0.660 | 0.705 | 0.661 | 0.706 |
| Sobel | Crop & Resize | 0.777 | 0.923 | 0.805 | 0.781 | **0.887** | 0.715 | **0.742** | **0.724** | 0.651 | 0.764 |
| Crop & Resize | Distort | **0.780** | **0.924** | **0.855** | **0.802** | 0.884 | **0.719** | 0.717 | 0.679 | **0.655** | **0.788** |

Table 17: Evaluation of $t_\theta$ linear probing on MIMIC-CXR with other Siamese representation learning strategies. Numbers are AUROC. DINO (global) refers to DINO with 2 global crops, and 0 local crops. DINO (local) refers to DINO with 2 global crops and 6 local crops.

| | | Evaluation Data | | |
|---|---|---|---|---|
| Framework | (Pre)Training Data | MIMIC-CXR | CheXpert | VinDr-CXR |
| SimSiam | MIMIC-CXR | 0.760 | 0.649 | 0.765 |
| SimCLR | MIMIC-CXR | 0.779 | 0.709 | 0.810 |
| MoCo V2 | MIMIC-CXR | 0.785 | 0.702 | 0.781 |
| DINO (global) | MIMIC-CXR | 0.720 | 0.606 | 0.683 |
| DINO (local) | MIMIC-CXR | 0.726 | 0.581 | 0.678 |
| FS (S) | MIMIC-CXR | 0.710 | 0.612 | 0.668 |
| FS (IN) | MIMIC-CXR | 0.742 | 0.630 | 0.710 |

Table 18: Expanded table for MIMIC-CXR $t_\theta$ to CheXpert transfer. Macro AUROC. Abbreviations: ZS: Zero-shot transfer, LP: Linear Probing, FT: Fine-tuning, FS: Fully Supervised, S: trained from scratch, IN: ImageNet, Chex: CheXpert, Mimic: MIMIC-CXR, VinDr: VinDr-CXR, Eval: Evaluation.

| Eval on CheXpert | | | | | Eval on MIMIC-CXR | | | | | Eval on VinDr-CXR | | | | |
|---|---|---|---|---|---|---|---|---|---|---|---|---|---|---|
| ZS | LP (Chex) | FT (Chex) | FS (S) (Chex) | FS (IN) (Chex) | ZS | LP (Chex) | FT (Chex) | FS (S) (Mimic) | FS (IN) (Mimic) | ZS | LP (Chex) | FT (Chex) | FS (S) (VinDr) | FS (IN) (VinDr) |
| 0.649 | 0.743 | 0.768 | 0.757 | **0.789** | 0.760 | 0.728 | **0.763** | 0.710 | 0.742 | 0.765 | 0.754 | **0.810** | 0.668 | 0.719 |

