# OpenReview forum: "Exploring Image Augmentations for Siamese Representation Learning with Chest X-Rays"
_MIDL.io/2023/Conference — MIDL 2023 Oral_

### Official Review · Reviewer_PUnq · 2023-01-27

**Confidence:** 4
**Preliminary Rating:** 5
**Recommendation:** Poster

**Summary:**

This papers examines image augmentation strategies for self-supervised Siamese representation learning, and the utility of the learned representation through linear probing, fine-tuning, zero-shot transfer, and data efficiency. Experiments were conducted on 3 datasets to allow evaluation on generalization and transfer of the learned representations.


**Strengths:**

- To investigate how common augmentation strategies successful with self-supervised learning in natural images may perform in medical images is important.

- The experiments, especially in the various strategies to evaluate the utility of the learned representations by linear probing, zero-short generalization, and transfer learning, are comprehensive and highly appreciated. The findings provide evidence on the use of such representation learning in medical images.

**Weaknesses:**

- A moderate concern is that, while the authors stated that the study investigates the effect of various augmentations on the learned representations, it is only the first part of the investigation that compares the effectiveness of various augmentation strategies. Once the “optimal strategy” is identified, it became the only augmentation strategies being evaluated for the rest of the experiments. For the latter, the results are then almost expected regarding the relative performance between the FS and using the learned representations. It’d would have been ideal if the effect of various augmentations, at least a subset of them, is evaluated for all the tasks considered.

**Deanonymize Review:**

no

**Paper Type:**

validation/application paper

**Questions To Address In The Rebuttal:**

Add comment and discussion on the concern above, especially whether the authors believe that the “optimality” of the data augmentation strategies may change depending on the tasks it is being evaluated on (e.g., zero-shot, fine-tuning, etc)

---

### Official Review · Reviewer_ZNsk · 2023-02-02

**Confidence:** 4
**Preliminary Rating:** 5
**Recommendation:** Poster

**Summary:**

The paper proposes to explore different image augmentations for self-supervised learning, specifically the SimCLR method, in the domain of Chest X-Ray. Results from experiments on three different Chest X-ray datasets and a combination of multiple different data augmentation techniques show that a SimCLR network trained with RandomResizeCropping followed by intensity distortion provides the best-learned features. Overall paper is well-motivated with an abundance of experiments.

**Strengths:**

* Introduction section and Literature Review section are really well written, with great coverage for both general-purpose self-supervised learning methods and medical image analysis-related self-supervised learning methods.
* Experiments on three different publicly available datasets are commendable.

**Weaknesses:**

* **Self-supervised learning for Chest X-rays** subsection in Section:2 mentions that the most closely related work to their study is MoCo-CXR (Sowirajan et al., 2021) and MICLe (Azizi et al., 2021). I think they are missing Azizi et al. 2022 [1] as it also uses SimCLR, similar to the study done in this work. They also utilized different augmentation techniques and evaluated SimCLR in out-of-distribution settings.
* Section 3.2 does not clarify how the dataset was divided into training and validation purposes. It mentions that the CheXpert dataset contains 224316 x-rays, from which 168660 were used as training and 22367 were used as validation. Does that mean the rest of the images were used for testing? If that is the case, it should be written in the paper.
* It is unclear how the data is divided between the unlabeled set used for training SimCLR and the labeled set used for the downstream classification task across all three datasets. Table 5 in the appendix does not contain all the necessary information. Also, some things in that table are unclear. For example, the summation of the percentage given in the brackets across a single column (ex., MIMIC-CXR Training) is more than 100%.
* **Section 4.2**: I think there is a typo here, as the authors refer to Table 4 while the results are reported in Table 1. While the VinDr-CXR results are not reported in Table 1 as written in the paper text. The paper text talks about $\textit{fracture}$ class, but in Table 1, it is written as RF. Authors may want to be consistent here. Similarly, $t_{\theta}$ should be replaced with Linear Prob in Table 1 to be consistent with other tables.
* In **Section 4.3**, authors write the following sentence "In zero-shot transfer to VinDr-CXR pathologies available in MIMIC-CXR, the $t_{\theta}$ representations achieve 0.767 AUROC, outperforming fully supervised VinDr-CXR networks by 0.099 and 0.057 AUROC when trained from scratch and ImageNet, respectively (Table 2)." These reported results are not from zero-shot transfers but are from Linear Probing.
* Similarly, in the same section, the authors mention the results of zero-shot transfer with the CheXpert dataset. No table is referred to here, and these results are not found in any of the tables in the paper.
* In **Section 4.4**, authors talk about $\textit{VinDr-Imbalanced}$ and $\textit{VinDr-Balanced}$ datasets. And mentions their performance. No table is referred to here, and these results are not found in any of the tables in the paper.
*  **Section 4.4**: The paper text talks about $\textit{tuberculosis}$ class, but in Table 3, it is written as TB. Consistency across these would be helpful.
* In **Section 4.5**, authors write that MIMIC-CXR self-supervised and CheXpert fine-tuned model gives 0.810 AUROC on the VinDr-CXR dataset. Can the authors explain why this performance is higher than the Fully supervised model (FN - IN)  on VinDR-CXR?
* Can authors give some clarification on why Linear Probing results are not reported in Table 3?
* Would the authors expect further performance improvement if more than 2 data augmentation techniques are combined?

[1] Azizi, S., Culp, L., Freyberg, J., Mustafa, B., Baur, S., Kornblith, S., Chen, T., MacWilliams, P., Mahdavi, S.S., Wulczyn, E. and Babenko, B., 2022. Robust and efficient medical imaging with self-supervision. arXiv preprint arXiv:2205.09723.

**Deanonymize Review:**

no

**Detailed Comments:**

* **Self-supervised learning** subsection in Section:2 has a wrong citation. Authors cite Zhang et al., 2016 for image rotation prediction. I think it should be Gidaris et al., 2018.
* At the start of Section 3 authors mention that they use SimSiam (Chen et al., 2020) network, while in Section 4.1, they say that their study was inspired based on SimCLR by Chen et al., 2020. The authors are using two different abbreviations for the same paper. Consistency for this would be good.
* Figure-2 caption should mention that results reported on the right side of the figure are based on linear probing.
* In some places, authors use 2-point precision for reporting AUROC, such as Section 4.2 and Figure 2, while in other places, they use three-point precision. Authors may want to use consistency here.

* In the future, authors may want to try other augmentation techniques, such as those proposed in [1], to learn more robust representation.
* Similarly, in the future, the authors may want to consider learned group augmentation techniques such as the one proposed in [2].

[1] Papakipos, Z. and Bitton, J., 2022. AugLy: Data augmentations for adversarial robustness. In Proceedings of the IEEE/CVF Conference on Computer Vision and Pattern Recognition (pp. 156-163).

[2] Wagner, D., Ferreira, F., Stoll, D., Schirrmeister, R.T., Müller, S. and Hutter, F., 2022. On the Importance of Hyperparameters and Data Augmentation for Self-Supervised Learning. arXiv preprint arXiv:2207.07875.

**Paper Type:**

both

**Questions To Address In The Rebuttal:**

Mainly all the points raised in the Weakness section. If the authors can incorporate some of the suggestions made in the "Detailed Comments" section, it would help to improve the clarity of the paper.

Edit: After the response by the authors during the rebuttal period, I am happy to change my score to Strong Accept.

---

### Official Review · Reviewer_HhTB · 2023-02-10

**Confidence:** 4
**Preliminary Rating:** 4
**Recommendation:** Poster

**Summary:**

This paper presents an experimental validation of self-supervised pre-training
using augmentation on three public chest x-ray datasets. The existing SimSiam
method is used for self-supervised pre-training using augmented versions of the
images, but the augmentations are varied with the rationale that the optimal
augmentations for medical images may be different from those for natural
images. Multiple experiments demonstrate the effectiveness of augmentation
based self-supervised learning methods over training from scratch. Further
comparisons investigate zero-shot transfer, linear probing (fine tuning only
last layer), and full fine-tuning.


**Strengths:**

The paper is clearly written and easy to understand. The experiments are very
thorough and the results will be of interest to the MIDL community as a benchmark
of how well augmentation based fine-tuning works for medical images, as opposed
to natural images. Multiple datasets large datasets are used, suggesting the results
are likely generalizable and reliable.



**Weaknesses:**

The novelty is limited as this is an experimental evaluation of an existing
method on medical datasets.

Limiting the experiments to only augmentation based self-supervised techniques
limits its utility - comparison to other methods such as DINO, MoCo, BYOL would
be useful. The authors should mention this in their conclusion: other pre-training
methods may perform better.

Since each training process is only performed once, some results may be due to
chance and sub-optimal initalization.

**Deanonymize Review:**

no

**Detailed Comments:**

Distortion used in Figure 2 but not defined. Is this the same as
contrast/brightness adjustments?

When discussing the zero shot transfer, there is a missed opportunity for a
simple additional comparison. Does the model trained via the t_theta strategy
on the MIMIC-CXR dataset perform better (zero-shot) on the other datasets than
the model trained from scratch *on MIMIC-CXR* performs (zero-shot) on the same
datasets? This would hint at whether the augmentation based pre-training gives
more robust/generalizable models.

The authors state that the improvement of t_theta over fully supervised
pre-training is consistent over all pathologies (section 4.4). But it appears
that for pleural effusion (PE) the performance of the fully supervised model
is actually significantly higher, according to table 3.

In Section 4.6, the authors state that "For CheXpert evaluation, we see that even
1% fine-tuning improves performance over zero-shot transfer by over 0.05 AUROC".
However this does not match my reading of Table 4, where the relevant numbers
seem to be 0.649 (zero shot) and 0.679 (1% fine tuning), which is a difference
of just 0.03...

It would be interesting to know whether combining more than two augmentations
continues to improve performance. Obviously running all combinations is not practical
but some further representative examples would be useul.

**Paper Type:**

validation/application paper

**Questions To Address In The Rebuttal:**

I am confused by the experiments in which both transformations are the identity
transform. In this case, the two network inputs are surely the same. And the
weights are the same, since they are shared. So both z_i and p_i will also
be the same, so how does the model learn anything? The authors should clarify
this and its significance.

---

### Meta-Review · Area_Chair_sTdf · 2023-02-24

**Recommendation:** Accept (Oral)
**Confidence:** 5

**Metareview:**

This paper explores image augmentation, an important aspect of training in our field, for self-supervised methods in x-rays.

This is a great example of what we want to see form the peer review process -- a great back and forth with feedback from the reviewers helping the paper. The authors went through extensive and professional rebuttal, which was highly appreciated by all. There's really not all that much to say except congratulations, and I hope that the authors make all the changes (I know there is a revision updated, but it's not easy to check all the changes in this timeframe).

One small thing from me is that it would be interesting to understand the extend of the generalization of the conclusions of this work to other domains of medical imaging beyond xrays. Obviously this is beyond the scope, but it would be super useful and impactful.